# Ask a Strong LLM Judge when Your Reward Model is Uncertain

**Zhenghao Xu**[1][*]   **Qin Lu**[2]   **Qingru Zhang**[1][*]   **Liang Qiu**[2]   **Ilgee Hong**[1][*]   **Changlong Yu**[2]

**Wenlin Yao**[2]   **Yao Liu**[2]   **Haoming Jiang**[2]   **Lihong Li**[2]   **Hyokun Yun**[2]   **Tuo Zhao**[1]

[1]Georgia Institute of Technology    [2]Amazon

## Abstract

Reward model (RM) plays a pivotal role in reinforcement learning with human feedback (RLHF) for aligning large language models (LLMs). However, classical RMs trained on human preferences are vulnerable to reward hacking and generalize poorly to out-of-distribution (OOD) inputs. By contrast, strong LLM judges equipped with reasoning capabilities demonstrate superior generalization, even without additional training, but incur significantly higher inference costs, limiting their applicability in online RLHF. In this work, we propose an uncertainty-based routing framework that efficiently complements a fast RM with a strong but costly LLM judge. Our approach formulates advantage estimation in policy gradient (PG) methods as pairwise preference classification, enabling principled uncertainty quantification to guide routing. Uncertain pairs are forwarded to the LLM judge, while confident ones are evaluated by the RM. Experiments on RM benchmarks demonstrate that our uncertainty-based routing strategy significantly outperforms random judge calling at the same cost, and downstream alignment results showcase its effectiveness in improving online RLHF. Our code is available at `https://github.com/zhenghaoxu-gatech/uncertainty-router`.

## 1   Introduction

Reinforcement learning with human feedback (RLHF) is a predominant approach for large language model (LLM) alignment and has shown great success in improving the capabilities of LLMs [7, 63, 44, 38, 3]. This approach formulates alignment as an RL problem in which the LLM, as the actor, is tuned to maximize a reward function that reflects human preference. This RL problem is then solved by policy gradient (PG) type of methods, including PPO [39], GRPO [43], and RLOO [1].

The reward function in RLHF is typically realized by a reward model (RM) learned from human-annotated preference data, assuming that human preference follows the Bradley-Terry (BT) model [5]. This *pointwise* RM assigns a scalar reward score $r(\boldsymbol{x}, \boldsymbol{y})$ to a response $\boldsymbol{y}$ measuring the quality of this response to the prompt $\boldsymbol{x}$, which estimates the ground truth reward underlying the BT model [7, 44]. A variant of this pointwise RM is the *pairwise* RM, or preference model (PM), which relax the BT assumption and assigns a scalar preference score $p(\boldsymbol{x}, \boldsymbol{y}_1, \boldsymbol{y}_2)$ to a pair of responses $\boldsymbol{y}_1$ and $\boldsymbol{y}_2$ given prompt $\boldsymbol{x}$, measuring the preference strength between the two responses [36, 46, 55, 61]. Both RMs are typically trained based on a pretrained LLM, concatenating the input as a single sequence and directly outputting a scalar reward/preference score. They are moderately fast and have been successfully integrated into the RLHF pipeline.

---

[*]Work done during internship at Amazon. Emails: {zhenghaoxu,tourzhao}@gatech.edu

39th Conference on Neural Information Processing Systems (NeurIPS 2025).

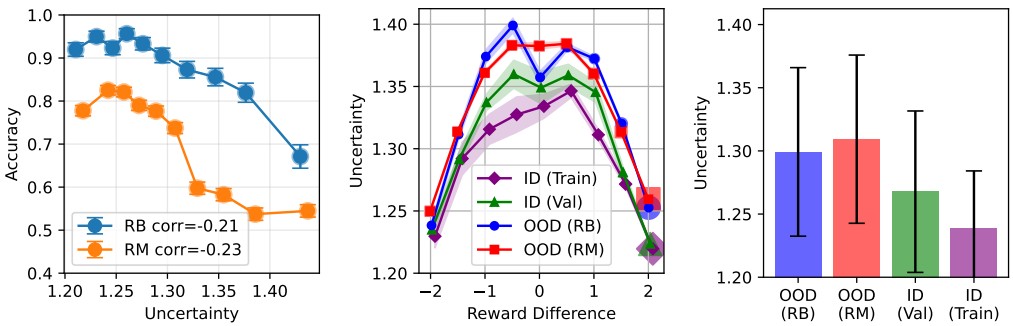

Figure 1: **Left**: Average accuracy of pairwise RM at uncertainty quantiles on RewardBench (RB) and RM-Bench (RM). Each scatter point corresponds to 10% of the data corresponding to a quantile. The accuracy shows a negative Spearman's rank correlation with uncertainty, with p-values less than $10^{-29}$ and $10^{-136}$ on RewardBench and RM-Bench, respectively, suggesting that uncertain pairs are more likely to get wrong rankings. **Middle**: Uncertainty gap between ID (HelpSteer2-Preference, train and validation sets) and OOD datasets (RewardBench and RM-Bench), where the uncertainty scores given by our RM range from 1. Uncertainty scores are averaged within 10 bins divided by reward difference. The marker size indicates the portion of data contained within the bin, and the shaded area represents the range within a standard deviation. **Right**: Overall averaged uncertainty scores. The uncertainty scores are consistently higher on OOD data.

However, RM could be vulnerable to hacking [2] and susceptible to spurious features such as specific styles [15]. For example, when tested on the hard subset of RM-Bench [31], which evaluates the RM's ability to distinguish subtle content changes and resistance to style biases, even the state-of-the-art models like Skywork-Reward-Llama-3.1-8B [27] struggle to achieve a higher accuracy (46.6%) than a random guess (50%). Because the RM is trained on rather limited human preference data, which cannot exhaustively cover all possible responses, the RM possesses a lot of epistemic uncertainty and falls short in making reliable predictions when facing out-of-distribution (OOD) data. As illustrated in Figure 1, the RM accuracy can drop significantly on uncertain OOD data. Therefore, RM is still far from a satisfactory objective.

Given the issue of standard RM, recent works turn to strong *generative* LLM judges for more reliable reward and preference annotations [62]. The LLM judge concatenates the input sequence with judge rubrics and autoregressively generates an output sequence containing the verdict, which can be extracted by simple pattern matching. By leveraging long chain-of-thought (CoT), a strong LLM judge can reason before giving a final answer, enabling inference time scaling for a more reliable return [53, 34, 58, 18]. For example, Deepseek-R1 [18] can achieve 78.9% accuracy on the hard subset of RM-Bench (see Table 4), outperforming traditional RMs by a huge amount.

Although LLM judges can make more accurate predictions, they are significantly more costly than RMs due to their reliance on autoregressive generation and long CoT. Consequently, their inference can take many times longer than that of standard scalar RMs, even with ample hardware and parallel execution. This high latency renders them a bottleneck in policy optimization, making their direct deployment in online RLHF pipelines intractable.

To address all these challenging issues of RM and LLM-as-a-judge, we propose an uncertainty-based routing framework to provide reliable reward signals at an affordable cost. We first quantify the uncertainty of RM predictions, and then use the uncertainty score as an indicator for routing. If the uncertainty is above a threshold, we recognize the data as an OOD sample and send it to the strong LLM judge for a more accurate verdict. If the uncertainty is low, then the sample is more likely to be in-distribution (ID), where the RM can provide a confident prediction at a fast speed, as no autoregressive decoding is required. Applying this uncertainty-based routing, we can complement standard RM with a strong LLM judge to improve OOD performance with lower cost, striking a balance between the two, and making it capable of enhancing the downstream online RLHF.

In particular, we use the pairwise PM instead of pointwise RM, because they can better capture human preference beyond the BT model [36, 46, 61]. Moreover, pointwise BT RM is indefinite, making it difficult to quantify its uncertainty from human preference data (more details in Section 2.3).

Contrarily, pairwise PM learning is a well-defined classification problem, and thus various principled uncertainty quantification methods can be applied, and we particularly use SNGP [28, 29] since it only requires a single model and inference once for each pair. While PM does not directly serve as an RL objective, it can be used to estimate the advantage and thus be applied to a class of PG methods, including GRPO [43] and RLOO [1].

We conduct experiments to demonstrate the efficacy of our uncertainty-based routing method. Firstly, we evaluate on RM benchmarks, showing that sending uncertain samples to a strong LLM judge can improve preference prediction accuracy without incurring too much cost. Then, we compare the uncertainty router with randomly routing the same number of samples to the judge. We evaluate their accuracy on reward benchmarks and apply them to downstream alignment. As illustrated in Figure 2 and detailed in Tables 2, 4 and 6, routing the uncertain samples can bring more improvement compared to random routing, showcasing the efficacy of using uncertainty as a routing indicator.

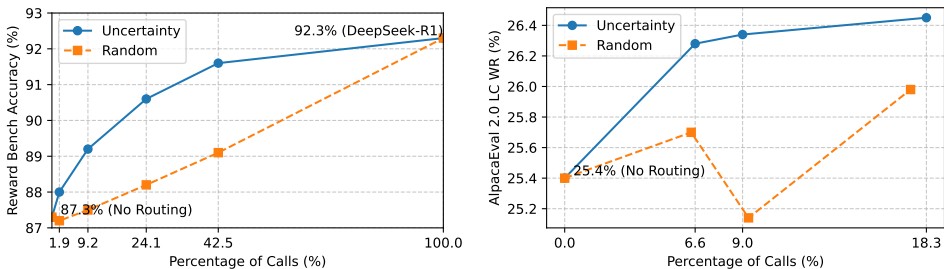

Figure 2: Uncertainty-based routing outperforms random routing with the same number of LLM judge calls on preference accuracy and downstream alignment. See Figure 3 for other benchmarks.

The paper is organized as follows: Section 2 introduces the preliminaries, Section 3 introduces our main methods, Section 4 provides experiments on reward benchmarks and downstream alignment, and Section 6 makes concluding remarks. Additional related work and experimental details are provided in the appendix.

## 2 Preliminaries

### 2.1 Reinforcement Learning with Human Feedback

Consider an LLM $\pi_{\boldsymbol{\theta}}$ with parameter $\boldsymbol{\theta}$, which takes a length-$L_{\text{in}}$ sequence $\boldsymbol{x} = [x_1, \ldots, x_{L_{\text{in}}}]$ as input and outputs a length-$L_{\text{out}}$ response $\boldsymbol{y} = [y_1, \ldots, y_{L_{\text{out}}}]$ sampled from its conditional probability distribution $\pi_{\boldsymbol{\theta}}(\cdot \mid \boldsymbol{x})$ over all possible output sequences. Reinforcement learning with human feedback (RLHF) aims to maximize the expected reward under the LLM policy $\pi_{\boldsymbol{\theta}}$, where the reward function is usually given by a reward model (RM) learned from human preference data. The preference data $\mathcal{D}_{\text{pref}} = \{(\boldsymbol{x}, \boldsymbol{y}_w, \boldsymbol{y}_l)\}$ are assumed to be following the Bradley-Terry (BT) model [5] with some unknown ground truth reward function $r^{\star}(\cdot, \cdot)$:

$$\mathbb{P}(\boldsymbol{y}_w \succ \boldsymbol{y}_l \mid \boldsymbol{x}) = \sigma(r^{\star}(\boldsymbol{x}, \boldsymbol{y}_w) - r^{\star}(\boldsymbol{x}, \boldsymbol{y}_l)), \qquad (1)$$

where $\boldsymbol{y}_w \succ \boldsymbol{y}_l$ denotes that response $\boldsymbol{y}_w$ is preferred over response $\boldsymbol{y}_l$ for prompt $\boldsymbol{x}$, and $\sigma(x) := 1/(1 + e^{-x})$ denotes the sigmoid function. A *pointwise* RM $r(\cdot, \cdot)$ can thus be trained with the following maximum likelihood estimation (MLE) loss:

$$\min_{r:\,(\boldsymbol{x}, \boldsymbol{y}) \mapsto \mathbb{R}} \mathcal{L}_{\text{pointRM}}(r) := \mathbb{E}_{(\boldsymbol{x}, \boldsymbol{y}_w, \boldsymbol{y}_l) \sim \mathcal{D}_{\text{pref}}}[-\log(\sigma(r(\boldsymbol{x}, \boldsymbol{y}_w) - r(\boldsymbol{x}, \boldsymbol{y}_l)))]. \qquad (2)$$

Given the RM, RLHF aims to solve the following RL problem over a prompt dataset $\mathcal{D}$:

$$\max_{\boldsymbol{\theta}} \mathcal{J}_{\text{policy}}(\boldsymbol{\theta}) := \mathbb{E}_{\boldsymbol{x} \sim \mathcal{D}, \boldsymbol{y} \sim \pi_{\boldsymbol{\theta}}(\cdot | \boldsymbol{x})}[r(\boldsymbol{x}, \boldsymbol{y})] - \beta \mathbb{E}_{\boldsymbol{x} \sim \mathcal{D}}[D_{\text{KL}}(\pi_{\boldsymbol{\theta}}(\cdot \mid \boldsymbol{x}) \parallel \pi_{\text{ref}}(\cdot \mid \boldsymbol{x}))], \quad (3)$$

where $\pi_{\text{ref}}(\cdot \mid \cdot)$ is a reference policy, $D_{\text{KL}}(\cdot \parallel \cdot)$ is the Kullback–Leibler (KL) divergence between two probability distributions, and $\beta > 0$ is the regularization factor. The problem (3) is usually solved via policy gradient (PG, [54, 45]) methods, including PPO [39], RLOO [1] and GRPO [43].

One can also train a *pairwise* RM, also called preference model (PM), to estimate the ground truth reward difference, $p(\boldsymbol{x}, \boldsymbol{y}_1, \boldsymbol{y}_2) \approx r^{\star}(\boldsymbol{x}, \boldsymbol{y}_1) - r^{\star}(\boldsymbol{x}, \boldsymbol{y}_2)$, which is equivalent to estimating the logits

in binary classification. The PM can be trained with the following loss:

$$\min_{p:\,(\boldsymbol{x},\boldsymbol{y}_1,\boldsymbol{y}_2)\mapsto\mathbb{R}} \mathcal{L}_{\mathrm{pairRM}}(p) := \mathbb{E}_{(\boldsymbol{x},\boldsymbol{y}_w,\boldsymbol{y}_l)\sim\mathcal{D}_{\mathrm{pref}}}[-\log(\sigma(p(\boldsymbol{x},\boldsymbol{y}_w,\boldsymbol{y}_l)))]. \tag{4}$$

If the data come from the BT model, then ideally minimizing (4) yields a PM consistent with the ground truth reward difference, which is sufficient for constructing advantage estimates for RLOO and GRPO. We provide more details when introducing our methods in Section 3.2.

## 2.2 LLM as a Judge

Powerful LLMs (e.g., GPT-4) are increasingly used as automated judges [4, 16, 62] to reduce costly human annotations. This AI feedback approach prompts an LLM judge to evaluate competing responses $(\boldsymbol{y}_1, \boldsymbol{y}_2)$ for an input $\boldsymbol{x}$, providing preference labels $(\boldsymbol{y}_w, \boldsymbol{y}_l)$ at scale. The quality of LLM judges benefits significantly from chain-of-thought (CoT) prompting [53] and recent progress in LLM reasoning capabilities, which incentivizes the LLM judge to generate a long reasoning path before giving a verdict, enhancing the reliability and transparency of evaluations [30].

Despite being cheaper than human annotation, LLM-as-a-judge still incurs significant computational cost due to the resource requirements for inference. Furthermore, high inference latency, especially when generating detailed reasoning, can slow down the process and present practical challenges to apply to online RLHF [51]. This necessitates the balance of feedback quality, speed and cost.

## 2.3 Uncertainty Quantification

Uncertainty quantification (UQ) aims to provide calibrated estimates of confidence associated with model predictions. In deep supervised learning, various UQ methods have been developed, particularly for classification tasks, including MC Dropout [13], Deep Ensembles [24], and methods focusing on distance-awareness in input or feature space, such as DUQ [47] and SNGP [28, 29]. In particular, SNGP combines spectral normalization with a Gaussian process layer for uncertainty estimation with distance awareness, which is useful for OOD detection without the multiple inference cost of MC dropouts or ensembles. For binary classification data, it outputs a logit that indicates the aleatoric uncertainty (irreducible due to the nature of data distribution), divided by a variance factor measuring the distance to the training set, which indicates the epistemic uncertainty (reducible by expanding data coverage).

Recent research has explored uncertainty quantification methods for RMs, such as LoRA ensembles [59] and last layer embedding [60]. While these approaches attempt to estimate uncertainty for pointwise RMs to enhance reliability, they face fundamental limitations. A key challenge is that pointwise RMs are inherently indefinite under the Bradley-Terry preference model - adding a prompt-dependent-only bias term yields the same preference distribution, making the UQ problem ill-defined. This ambiguity complicates the interpretation of uncertainty estimates from ensemble or kernel/GP methods. For instance, high variance in RM ensemble predictions could indicate either genuine uncertainty or simply convergence of RMs to different but equivalent solutions. While high-quality scalar ratings of individual responses could make pointwise RM learning a regression problem where UQ is well-posed [33, 11], such data is typically less available than human preference data [50, 52]. In this work, we specifically address uncertainty quantification for pairwise RM (PM) trained on human preference data that are more widely used in the RLHF literature [7, 63, 44, 38, 3].

## 3 Method

We aim to address the issue of poor generalization of RM on OOD data in order to get better downstream alignment performance. Instead of improving the RM itself, we investigate complementing the RM with a strong external LLM judge, which provides more reliable preference feedback. However, such an external LLM judge incurs high inference cost and latency, making it unrealistic, given the limited computational budget, to evaluate every response generated by the actor model during online RLHF. Therefore, we need to specify a strategy to switch between cheap but weak RM and the strong but expensive LLM judge in order to maximize the gain from the limited number of judge calls. To achieve this, we propose a routing framework based on RM uncertainty quantification (UQ), sending the data that RM is uncertain about to the LLM judge for further evaluation.

### 3.1 Reward Difference Estimation with Uncertainty Quantification

We first train a pairwise RM (PM), $p(\cdot, \cdot, \cdot)$, to estimate the ground-truth reward difference with uncertainty quantification using preference data $\mathcal{D}_{\mathrm{pref}} = \{(\boldsymbol{x}, \boldsymbol{y}_w, \boldsymbol{y}_l)\}$. In particular, we concatenate the prompt and two responses into a single input sequence using a chat template (details in Appendix A). We take the hidden states at the last token of the concatenated input as an embedding vector and add a head outputting the classification logit $p(\boldsymbol{x}, \boldsymbol{y}_w, \boldsymbol{y}_l)$ used in (4).

To mitigate the position bias that the two responses are concatenated only in the chosen-rejected order, we swap the positions and flip the labels, so that the augmented dataset $\overline{\mathcal{D}}_{\mathrm{pref}}$ contains both $(\boldsymbol{x}, \boldsymbol{y}_w, \boldsymbol{y}_l, 1)$ and $(\boldsymbol{x}, \boldsymbol{y}_l, \boldsymbol{y}_w, 0)$. We minimize the following classification loss to get a PM:

$$\min_{p\colon (\boldsymbol{x}, \boldsymbol{y}_1, \boldsymbol{y}_2) \mapsto \mathbb{R}} \mathcal{L}_{\mathrm{pref}}(p) := \mathbb{E}_{(\boldsymbol{x}, \boldsymbol{y}_1, \boldsymbol{y}_2, z) \sim \overline{\mathcal{D}}_{\mathrm{pref}}} \big[ -z \cdot \log(\sigma(p(\boldsymbol{x}, \boldsymbol{y}_1, \boldsymbol{y}_2)))$$
$$- (1-z) \cdot \log(\sigma(p(\boldsymbol{x}, \boldsymbol{y}_2, \boldsymbol{y}_1)))\big]. \tag{5}$$

Given the well-posedness of the classification problem (5), we can apply principled uncertainty quantification methods to detect OOD data. In particular, we apply the spectral-normalized Gaussian process (SNGP, [28, 29]) method, as it only requires a single model and infers once for each pair. When applying SNGP to LLM-based PM, we add spectral normalization [35] to the linear output layer in transformer blocks, take the final hidden states at the last token $\boldsymbol{h} = \boldsymbol{h}(\boldsymbol{x}, \boldsymbol{y}_1, \boldsymbol{y}_2) \in \mathbb{R}^{D_h}$ and pass it to a Gaussian process (GP) layer (approximated by random features) to get the logit $g(\boldsymbol{h})$ that corresponds to the reward difference:

$$g(\boldsymbol{h}) = \boldsymbol{\phi}(\boldsymbol{h})^\top \boldsymbol{\beta}, \quad \boldsymbol{\phi}(\boldsymbol{h}) = \sqrt{2\sigma_k^2/D_r} \cdot \cos(\boldsymbol{W}\boldsymbol{h} + \boldsymbol{b}) \tag{6}$$

where $\boldsymbol{\beta} \in \mathbb{R}^{D_r}$, $D_r$ is the number of random features, $\sigma_k^2$ is the kernel amplitude, $\boldsymbol{W} \in \mathbb{R}^{D_r \times D_h}$ is a fixed matrix with its entries i.i.d. sampled from standard Gaussian $\mathcal{N}(0, 1)$, and $\boldsymbol{b} \in \mathbb{R}^{D_r}$ is a fixed vector with its entries i.i.d. sampled from uniform distribution $\mathrm{Unif}(0, 2\pi)$.

During training, we plug $p(\boldsymbol{x}, \boldsymbol{y}_1, \boldsymbol{y}_2) = g(\boldsymbol{h}(\boldsymbol{x}, \boldsymbol{y}_1, \boldsymbol{y}_2))$ into (5) and apply gradient methods to update all hidden weights, except the fixed weights in the GP layer, i.e., $\boldsymbol{W}$ and $\boldsymbol{b}$. After training completes, we add an additional epoch to compute the posterior covariance matrix

$$\boldsymbol{\Sigma} = \mathrm{inv}(\boldsymbol{\Sigma}^{-1}), \quad \boldsymbol{\Sigma}^{-1} = \tau \boldsymbol{I} + \sum_{i=1}^{N} \sigma(p_i)(1 - \sigma(p_i))\boldsymbol{\phi}_i \boldsymbol{\phi}_i^\top, \tag{7}$$

where $p_i = p(\boldsymbol{x}^{(i)}, \boldsymbol{y}_1^{(i)}, \boldsymbol{y}_2^{(i)})$ and $\boldsymbol{\phi}_i = \boldsymbol{\phi}(\boldsymbol{x}^{(i)}, \boldsymbol{y}_1^{(i)}, \boldsymbol{y}_2^{(i)})$.

During inference, we compute the reward difference $p$ and uncertainty $u$ as

$$p(\boldsymbol{x}, \boldsymbol{y}_1, \boldsymbol{y}_2) = \frac{g(\boldsymbol{h})}{u(\boldsymbol{x}, \boldsymbol{y}_1, \boldsymbol{y}_2)}, \quad u(\boldsymbol{x}, \boldsymbol{y}_1, \boldsymbol{y}_2) = \sqrt{1 + \lambda \cdot \boldsymbol{\phi}(\boldsymbol{h})^\top \boldsymbol{\Sigma} \boldsymbol{\phi}(\boldsymbol{h})}, \tag{8}$$

where $g(\boldsymbol{h})$ and $\boldsymbol{\phi}(\boldsymbol{h})$ are defined in (6) and $\boldsymbol{\Sigma}$ is defined in (7), $\lambda$ is a scaling factor.

In this SNGP-PM, the logit $g$ quantifies the aleatoric uncertainty from the BT model, and the variance-induced uncertainty $u$ quantifies the epistemic uncertainty due to limited training data. The aleatoric uncertainty is not reducible as it is inherent in human preference; thus, applying an LLM judge to the aleatoric uncertain samples may not bring much improvement. On the other hand, epistemic uncertainty is reducible; thus, a strong LLM judge with good generalization may help improve the prediction on epistemic uncertain samples. Therefore, we use $u$ as our uncertainty quantifier.

**Remark 1** *As mentioned in Section 2.3, we consider PM instead of pointwise RM for preference data under the BT assumption because the uncertainty quantification problem is not well-posed for the latter. Consider a pointwise RM $r(\boldsymbol{x}, \boldsymbol{y})$, it is consistent with any other RM in the form of $r(\boldsymbol{x}, \boldsymbol{y}) + s(\boldsymbol{x})$ under the BT model, and we cannot guarantee which one is returned by minimizing (2) even with infinite data. Therefore, it is difficult to assign a prior distribution on the pointwise RM, which is crucial for uncertainty quantification. In contrast, PM is well defined within the data support and is unique in the population sense. Therefore, one can measure the distance from the data to the support of the training set for epistemic uncertainty quantification.*

## 3.2 Advantage Estimator from Reward Differences under Uncertainty-Based Routing

When serving the PM, we set a threshold and use the uncertainty in (8) to route to a strong LLM judge: if the uncertainty is below the threshold, we directly use the prediction from the PM; if the uncertainty is above the threshold, we call the LLM judge and use its prediction in turn. The estimated pairwise reward differences are then used to construct the advantage values, enabling downstream RLHF with a class of policy gradient (PG) methods, including GRPO [18] and RLOO [1].

More precisely, the (stochastic) policy gradient is computed by taking the gradient of policy loss:

$$\mathcal{L}_{\text{policy}}(\boldsymbol{\theta}) = -\frac{1}{B}\sum_{i=1}^{B} A_i \log \pi_{\boldsymbol{\theta}}(\boldsymbol{y}_i \mid \boldsymbol{x}_i), \tag{9}$$

where $\{\boldsymbol{x}_i\}_{i=1}^{B}$ is a batch of prompts, $\boldsymbol{y}_i \sim \pi_{\boldsymbol{\theta}}(\cdot \mid \boldsymbol{x}_i)$, and $A_i = r(\boldsymbol{x}_i, \boldsymbol{y}_i) - b(\boldsymbol{x}_i)$ is the estimated advantage of response $\boldsymbol{y}_i$ conditioned on prompt $\boldsymbol{x}_i$ compared to a baseline $b(\boldsymbol{x}_i)$ that only depends on the prompt $\boldsymbol{x}_i$. To reduce the variance, the baseline is usually set as the value function $b(\boldsymbol{x}) = \mathbb{E}_{\boldsymbol{y}\sim\pi_{\boldsymbol{\theta}}(\cdot|\boldsymbol{x})}[r(\boldsymbol{x},\boldsymbol{y})]$, which is approximated by a critic model as in PPO [39], or by Monte-Carlo (MC) samples as in RLOO [1] and GRPO [43]. For simplicity, we consider RLOO, which generates a group of responses for each prompt and uses the leave-one-out average to estimate the baseline and advantages. Suppose $\{\boldsymbol{y}_i\}_{j=1}^{K}$ are $K$ responses generated from $\pi_{\boldsymbol{\theta}}(\cdot \mid \boldsymbol{x})$, then the advantage for this group corresponding to prompt $\boldsymbol{x}$ is given by

$$A_i = r(\boldsymbol{x}, \boldsymbol{y}_i) - \frac{1}{K-1}\sum_{j\neq i} r(\boldsymbol{x}, \boldsymbol{y}_j) = \frac{1}{K-1}\sum_{j\neq i}(r(\boldsymbol{x}, \boldsymbol{y}_i) - r(\boldsymbol{x}, \boldsymbol{y}_j)). \tag{10}$$

In view of (10), estimating the advantage only requires reward differences between responses within each group, and thus our SNGP-PM with an uncertainty router is applicable. When the SNGP-PM is certain about the comparison, we use its predicted reward difference directly. When the SNGP-PM is uncertain, we call a strong LLM judge, assuming it can produce reliable feedback on the comparison.

We restrict the return to be one of the three labels, indicating that $\boldsymbol{y}_i$ is better, $\boldsymbol{y}_j$ is better, or they are tied (see Appendix A for details). Given the working assumption that the external judge is strong, we assign a high confidence score (corresponding to near 1 or 0 probability) in case it predicts that one of the responses is better, and assign low confidence score (corresponding to $1/2$ probability) in case it predicts tied which indicates the occurrence of aleatoric uncertainty. The reward differences corresponding to the confidence scores can be obtained by applying the inverse of the sigmoid function. Combining the two sources of reward difference via the uncertainty router, our serving PM makes the following prediction on tuple $(\boldsymbol{x}, \boldsymbol{y}_i, \boldsymbol{y}_j)$:

$$\widetilde{p}(\boldsymbol{x}, \boldsymbol{y}_i, \boldsymbol{y}_j) = \begin{cases} p(\boldsymbol{x}, \boldsymbol{y}_i, \boldsymbol{y}_j), & u(\boldsymbol{x}, \boldsymbol{y}_i, \boldsymbol{y}_j) \leq \overline{u}, \\ J(\boldsymbol{x}, \boldsymbol{y}_i, \boldsymbol{y}_j), & u(\boldsymbol{x}, \boldsymbol{y}_i, \boldsymbol{y}_j) > \overline{u}, \end{cases} \qquad J(\boldsymbol{x}, \boldsymbol{y}_i, \boldsymbol{y}_j) = \begin{cases} \sigma^{-1}(1-\epsilon), & \boldsymbol{y}_i \overset{J}{\succ} \boldsymbol{y}_j, \\ \sigma^{-1}(\epsilon), & \boldsymbol{y}_i \overset{J}{\prec} \boldsymbol{y}_j, \\ \sigma^{-1}(1/2), & \boldsymbol{y}_i \overset{J}{\sim} \boldsymbol{y}_j, \end{cases}$$

where $\overline{u}$ is the routing threshold, $0 < \epsilon \ll 1$, and $\overset{J}{\succ}, \overset{J}{\prec}, \overset{J}{\sim}$ denote the verdicts from the judge.

Using the reward differences, we compute the advantage estimate (10) and plug it into (9). Then, adding the KL regularization yields the policy loss associated with the prompt $\boldsymbol{x}$ for an RLOO step.

$$\mathcal{L}_{\text{RLOO}}(\boldsymbol{\theta}) = -\frac{1}{K(K-1)}\sum_{i=1}^{K}\sum_{j\neq i}\widetilde{p}(\boldsymbol{x}, \boldsymbol{y}_i, \boldsymbol{y}_j)\log\pi_{\boldsymbol{\theta}}(\boldsymbol{y}_i \mid \boldsymbol{x})$$
$$+ \beta D_{\text{KL}}(\pi_{\boldsymbol{\theta}}(\cdot \mid \boldsymbol{x}) \parallel \pi_{\text{ref}}(\cdot \mid \boldsymbol{x})). \tag{11}$$

The downstream RLHF is then performed by iteratively sampling a batch of prompts, generating responses from the policy model, and updating the weights by taking a gradient step on the loss (11).

**Remark 2** *The PM-based RLOO loss (11) has a connection with Nash learning with human feedback or self-play RLHF [36, 46, 57, 42, 6, 55, 61]. These works formulate the alignment problem as a minimax game instead of an RL problem as (3). In this work, we still follow the RL framework and use PM only to construct advantage estimates. Our work focuses on efficiently complementing the PM with LLM-as-a-judge, instead of improving the downstream alignment method itself.*

# 4 Experiments

In this section, we present our experiments that examine the performance of SNGP-PM with an uncertainty router to the LLM judge and its benefit to downstream alignment.

## 4.1 Experiment Setup

**Models.** For both pairwise RM (PM) training and downstream alignment, we use Llama-3.1-8B-Instruct [17] as our base model. When serving PM with an uncertainty router, we use the DeepSeek-R1 [18] model as a judge, as it already achieves high accuracy on reward benchmarks without specific fine-tuning (see Tables 2 and 4).

**Datasets.** For PM training, we use the HelpSteer2-Preference dataset [51], which consists of 7,118 high-quality preference pairs with 6,766 training data pairs and 352 validation data pairs. For downstream alignment, we use a subset (the first 33%) of the prompt from the Ultrafeedback dataset [8], which consists of about 20k prompts covering various domains including instruction following, truthfulness, honesty, and helpfulness.

**Benchmarks.** For RM evaluation, we use RewardBench [25] and RM-Bench [31] datasets. RewardBench contains 2,985 preference pairs measuring the RM's capabilities over the categories of chat, chat hard, safety, and reasoning. RM-Bench contains 1,327 prompts, each associated with 3 chosen responses and 3 rejected responses, consisting of 11,943 pairwise comparisons in total. The responses in RM-Bench are constructed to amplify the style bias, making it a hard benchmark for RM to accurately make correct predictions. We report the accuracy of distinguishing chosen and rejected responses.

For downstream aligned policy models, we evaluate their performance on three widely adopted open-ended instruction following benchmarks: Arena-Hard-v0.1 [26], AlpacaEval 2.0 [12], and MT-Bench [62]. These benchmarks ask the model to generate answers to a wide range of open-ended questions and use strong judge models to assess the quality of the response. We follow each benchmark's evaluation protocol and report corresponding scores. For Arena-Hard-v0.1, we report the win rate (WR). For AlpacaEval 2.0, we report the WR and length-controlled (LC) WR. For MT-Bench, we report the scores on two turns and their average. More details are provided in Appendix C.3.

**Baseline.** To show that SNGP uncertainty quantification would not affect the accuracy, we train a standard PM without the GP head as a baseline. To validate the efficacy of our *uncertainty-based routing* approach, we experiment with different uncertainty thresholds and compare with *random routing*. When random routing is applied, we still use the same uncertainty threshold, but only for counting the number of required calls within the batch. We then randomly sample the indices and call DeepSeek-R1 on those pairs.

## 4.2 Uncertainty-based Routing Improves OOD Generalization

We train a PM with SNGP as specified in Section 3.1 based on Llama-3.1-8B-Instruct and HelpSteer2-Preference dataset. We augment the data by swapping the two responses for each prompt and apply the message format in Appendix A to construct the actual dataset used for PM training. The HelpSteer2-Preference dataset contains a preference strength $s \in \{1, 2, 3\}$ associated with each tuple, so we use the following scaled BT loss as suggested in [51]:

$$\mathcal{L}_{\text{scaled}}(p) := -\mathbb{E}_{(\boldsymbol{x}, \boldsymbol{y}_1, \boldsymbol{y}_2, z, s) \sim \overline{\mathcal{D}}_{\text{pref}}}[s \cdot z \cdot \log(\sigma(p(\boldsymbol{x}, \boldsymbol{y}_1, \boldsymbol{y}_2))) + s \cdot (1 - z) \cdot \log(\sigma(p(\boldsymbol{x}, \boldsymbol{y}_2, \boldsymbol{y}_1)))].$$

We train for 2 epochs to prevent overfitting, and use the third epoch to compute the covariance matrix used for SNGP uncertainty estimation, during which the weights are frozen. For the baseline, we replace the GP layer with a simple linear head and train a standard PM using the same data and loss function for 2 epochs. More details are provided in Appendix C.1.

The standard PM and SNGP-PM are evaluated on the HelpSteer2-Preference validation set, Reward-Bench and RM-Bench, and the results are provided in Table 1. As illustrated, the two models perform comparably with less than 1% overall accuracy difference, suggesting that the additional uncertainty quantification component does not introduce significant overhead to prediction accuracy.

We then evaluate our uncertainty routing strategy on RewardBench and RM-Bench. To mitigate position bias during inference, for each tuple $(\boldsymbol{x}, \boldsymbol{y}_w, \boldsymbol{y}_l)$ of prompt, chosen and rejected responses, we use $\frac{p(\boldsymbol{x}, \boldsymbol{y}_w, \boldsymbol{y}_l) - p(\boldsymbol{x}, \boldsymbol{y}_l, \boldsymbol{y}_w)}{2}$ as the predicted reward difference and $\frac{u(\boldsymbol{x}, \boldsymbol{y}_w, \boldsymbol{y}_l) + u(\boldsymbol{x}, \boldsymbol{y}_l, \boldsymbol{y}_w)}{2}$ as the uncertainty. We set the threshold in $\{10.0, 1.45, 1.40, 1.35, 1.30\}$, and send the tuples whose uncertainty

Table 1: Comparison of standard preference model and SNGP-PM on HelpSteer2-Preference validation set, Reward Bench, and RM-Bench. The performance of two models are comparable.

| Model | Validation | Reward Bench | | | | | RM Bench | | | | |
|---|---|---|---|---|---|---|---|---|---|---|---|
| | acc | avg. | chat | chat hard | safety | reasoning | avg. | chat | math | code | safety |
| PM | 0.801 | 0.877 | 0.964 | 0.731 | 0.894 | 0.918 | 0.687 | 0.670 | 0.605 | 0.551 | 0.923 |
| SNGP-PM | 0.793 | 0.873 | 0.958 | 0.738 | 0.894 | 0.900 | 0.680 | 0.671 | 0.595 | 0.542 | 0.912 |

is beyond the threshold to the DeepSeek-R1 judge using the template specified in Appendix A. The sign of the final preference indicates the correctness of the prediction. For the baseline, we choose random routing that routes exactly the same number of tuples to DeepSeek-R1 but in a random way.

The prediction accuracies on RewardBench and RM-Bench are presented in Tables 2 and 4, respectively. From the tables, it is shown that calling strong LLM judges improves the RM accuracy, especially on the hard domains where PM (no routing) performs poorly, such as chat hard and reasoning sections in RewardBench, and math and coding in RM Bench. Moreover, the threshold routing approach is significantly more efficient than random routing, achieving higher accuracy gains with the same amount of total judge calls.

Table 2: Performance comparison on RewardBench with different routing strategies. The thresholds are chosen in $\{10.0, 1.45, 1.40, 1.35, 1.30\}$.

| Routing | Num of Calls | Reward Bench (%) | | | | |
|---|---|---|---|---|---|---|
| | | chat | chat hard | safety | reasoning | avg. (vs rand) |
| No routing | 0 | 95.8 | 73.8 | 89.4 | 90.0 | 87.3 |
| Uncertainty | 58 (1.9%) | 96.1 | 74.8 | 89.5 | 91.7 | **88.0 (+0.8)** |
| | 274 (9.2%) | 96.4 | 76.8 | 89.8 | 93.7 | **89.2 (+1.7)** |
| | 719 (24.1%) | 96.9 | 80.3 | 89.8 | 95.4 | **90.6 (+2.4)** |
| | 1270 (42.5%) | 98.3 | 81.2 | 90.0 | 97.0 | **91.6 (+2.5)** |
| Random | 58 (1.9%) | 95.5 | 74.0 | 89.4 | 89.9 | 87.2 |
| | 274 (9.2%) | 96.4 | 73.7 | 89.5 | 90.4 | 87.5 |
| | 719 (24.1%) | 95.0 | 75.9 | 90.2 | 91.5 | 88.2 |
| | 1270 (42.5%) | 95.5 | 77.5 | 91.6 | 91.9 | 89.1 |
| DeepSeek-R1 | 100% | 95.5 | 85.8 | 91.1 | 96.9 | 92.3 |

Table 3: Computational costs for different routing strategies on RewardBench.

| Uncertainty Threshold | 10 | 1.45 | 1.4 | 1.35 | 1.3 | <1 |
|---|---|---|---|---|---|---|
| Num of Calls (Ratio) | 0% | 1.9% | 9.2% | 24.1% | 42.5% | 100% |
| Inference Time (s) - Uncertainty | 107 | 156 | 245 | 540 | 609 | 1113 |
| Inference Time (s) - Random | 107 | 149 | 208 | 361 | 541 | 1113 |

Since hard instances may take more inference time from the LLM judge, we further record the wall clock time running the evaluations and compare the performance of uncertainty-based and random routing with the same amount of inference time. We run evaluations on 4 NVIDIA-A100 GPUs in parallel, each processing 25% of the comparisons with an SNGP-PM. We then gather all routed instances and send them to the remote-hosted DeepSeek-R1 judge in parallel, which can process 200 requests per minute. As shown in Tables 3 and 5, uncertainty-based routing requires more inference time than random routing, suggesting that the uncertain instances are indeed harder. Nevertheless, the uncertainty-based routing strategy achieves higher accuracy with less time. These experiments validate the efficacy of our uncertainty-based routing approach.

### 4.3 Uncertainty-based Routing Improves Downstream Alignment

We then experiment on downstream alignment. We apply RLOO (11) for online RLHF. For each group of $K$-responses to the same prompt, we construct a preference matrix $P \in \mathbb{R}^{K \times K}$ to estimate the advantages. We send $K(K-1)$ ordered pairs to SNGP-PM and get $P_{i,j} = p(\boldsymbol{x}, \boldsymbol{y}_i, \boldsymbol{y}_j)$ and

Table 4: Performance comparison on RM-Bench with different routing strategies. The thresholds are chosen in $\{10.0, 1.45, 1.40, 1.35, 1.30\}$.

| Routing | Num of Calls | RM Bench (%) | | | | | | | |
|---|---|---|---|---|---|---|---|---|---|
| | | chat | math | code | safety | easy | normal | hard | avg. (vs rand) |
| No routing | 0 | 67.1 | 59.5 | 54.2 | 91.2 | 87.2 | 72.0 | 44.9 | 68.0 |
| Uncertainty | 242 (2.0%) | 68.7 | 60.0 | 54.7 | 91.4 | 87.4 | 73.3 | 45.5 | **68.7 (+0.2)** |
| | 1285 (10.7%) | 69.6 | 64.9 | 59.7 | 92.0 | 89.1 | 76.3 | 49.2 | **71.6 (+1.6)** |
| | 3188 (26.7%) | 71.3 | 73.8 | 68.7 | 92.6 | 91.4 | 81.5 | 56.9 | **76.6 (+3.1)** |
| | 5270 (44.1%) | 73.2 | 83.5 | 78.3 | 92.7 | 93.6 | 86.9 | 65.3 | **81.9 (+4.8)** |
| Random | 242 (2.0%) | 67.5 | 60.2 | 54.9 | 91.2 | 87.4 | 72.4 | 45.6 | 68.5 |
| | 1285 (10.7%) | 68.0 | 63.6 | 57.2 | 91.3 | 88.0 | 74.1 | 48.0 | 70.0 |
| | 3188 (26.7%) | 69.6 | 69.3 | 63.6 | 91.3 | 89.4 | 77.0 | 54.0 | 73.5 |
| | 5270 (44.1%) | 70.1 | 76.8 | 69.9 | 91.6 | 91.0 | 81.3 | 59.1 | 77.1 |
| DeepSeek-R1 | 100% | 76.8 | 95.7 | 87.8 | 92.0 | 94.0 | 91.3 | 78.9 | 88.1 |

Table 5: Computational costs for different routing strategies on RM-Bench.

| Trigger Threshold | 10 | 1.45 | 1.4 | 1.35 | 1.3 | <1 |
|---|---|---|---|---|---|---|
| Num of Calls (Ratio) | 0% | 2.0% | 10.7% | 26.7% | 44.1% | 100% |
| Inference Time (s) - Uncertainty | 518 | 632 | 1113 | 2200 | 3007 | 5642 |
| Inference Time (s) - Random | 518 | 625 | 1093 | 1979 | 2615 | 5642 |

$U_{i,j} = u(\boldsymbol{x}, \boldsymbol{y}_i, \boldsymbol{y}_j)$. We let $P \leftarrow \frac{P - P^\top}{2}$ and $U \leftarrow \frac{U + U^\top}{2}$ to enforce an anti-symmetric preference matrix and a symmetric uncertainty matrix. We then follow Section 3.2 to get feedback from the DeepSeek-R1 judge when uncertainty is above the threshold and compute the advantages accordingly.

Given resource constraints, we set $K = 4$ and train on the first 33% of the Ultrafeedback prompts for 1 epoch. We set the threshold in $\{10.0, 1.35, 1.30, 1.20\}$ and compare the uncertainty-based and random routers. More training and evaluation details are provided in Appendices C.2 and C.3. As shown in Table 6, complementing the PM with DeepSeek-R1 as a judge during RLHF brings improvement to downstream policy performance with a small portion of calls. Moreover, uncertainty-based routing in general exhibits higher improvement, showcasing the efficacy of our routing strategy.

Table 6: Performance of downstream models trained with different routing strategies and thresholds. For random routing, we use the same threshold for counting but send samples to the judge randomly.

| Model | Num of Calls | Arena-Hard (%) | AlpacaEval 2.0 (%) | | MT-Bench | | |
|---|---|---|---|---|---|---|---|
| | | v0.1 WR | LC WR | WR | Turn 1 | Turn 2 | Avg |
| Base model | - | 24.5 | 22.31 | 23.63 | 7.98 | 6.80 | 7.47 |
| No routing (10.0) | 0 | 28.1 | 25.40 | 27.35 | **8.19** | 6.98 | 7.65 |
| Uncertainty (1.35) | 7668 (6.6%) | 28.9 | 26.28 | **28.97** | 8.05 | 7.19 | 7.65 |
| Uncertainty (1.30) | 10522 (9.0%) | 28.9 | 26.34 | 28.53 | 8.03 | 7.13 | 7.63 |
| Uncertainty (1.20) | 21363 (18.3%) | **29.8** | **26.45** | 28.91 | 7.95 | **7.40** | **7.71** |
| Random (1.35) | 7523 (6.4%) | 26.5 | 25.70 | 28.55 | 8.09 | 7.20 | **7.71** |
| Random (1.30) | 10854 (9.3%) | 27.7 | 25.14 | 28.29 | 7.93 | 6.74 | 7.41 |
| Random (1.20) | 20474 (17.5%) | 28.5 | 25.98 | 28.51 | 8.00 | 6.62 | 7.45 |

## 5 Additional Related Work

**Pointwise and pairwise reward model.** Classical pointwise reward models (RM) are typically learned from human preference data [3, 8] under the Bradley-Terry (BT) model [5], which consists of a main component in RLHF [7, 63, 44, 38]. These methods train the pointwise RM by performing maximum likelihood estimation (MLE) on the preference data and could lead to RMs that lack calibration across different prompts [56]. When absolute rating data is available, where there are

attribute scores assigned to individual responses, the pointwise RM can also be trained via regression [10, 49]. However, high-quality rating data is less abundant than preference data, as fine-grained scalar scores typically require more human labor and precise rating rubrics for reliable annotations [23, 50, 52]. Recently, some works have relaxed the BT assumption and explored using pairwise RM, also called preference model (PM), to model general human preference [61]. Such a PM is trained by classification on human preference data and can serve as an evaluator or preference annotator [22, 32]. It can also serve as the alignment objective in Nash learning with human feedback [36, 46, 57, 42, 6] or self-play policy optimization [55, 61, 56]. While formulated differently, these alignment methods have a deep connection with standard policy gradient methods with baseline estimated via group averaged reward [48], including RLOO [1] and GRPO [43].

**Connection to active learning.** Our uncertainty-based routing framework is conceptually aligned with active learning, which seeks to improve data efficiency by strategically selecting the most informative instances for labeling by an oracle [40]. In our work, the strong LLM judge acts as the oracle, and our uncertainty score serves as the acquisition function to identify the most uncertain pairs of responses. Classic query strategies in active learning often rely on model uncertainty, such as selecting the least confident predictions or using a committee of models to find contentious examples [41], which have been extended to deep learning through Bayesian methods [14]. While active learning has been explored for learning reward/preference functions from limited interaction in the RL context [19, 9], our primary contribution is the application of this principle to a direct online RLHF setting. Here, the feedback from the oracle is used immediately to construct advantage estimates for policy gradient updates, and feeding these newly labeled high-uncertainty pairs back into the reward model for continued training is a natural extension that would constitute a full active learning cycle.

## 6 Conclusion

In this work, we propose an uncertainty-based routing strategy to complement cheap but poorly generalized RM with more reliable but expensive LLM-as-a-judge efficiently for RLHF. Experiments demonstrate improvement on reward benchmarks and downstream alignment, and our uncertainty-based routing strategy outperforms random routing, showcasing the efficacy of our method, which is measured by the number of judge calls. While SNGP does not significantly increase the computational cost, a future analysis using iso-flops could provide a more fine-grained measure of computational cost by also accounting for the UQ method's overhead.

The current work directly uses DeepSeek-R1 as a judge, which is not specifically fine-tuned for the judge task and requires huge computational resources to host. One can potentially replace it with a smaller-scale generative RM [34] to further improve the judge quality and inference efficiency. Another future direction could be quantifying the hardness of the sample and the uncertainty of the LLM judge and enabling a hierarchical routing strategy, potentially unifying the RM under the generative paradigm and allocating inference time budget more efficiently. A related avenue is exploring alternative UQ methods to optimize the trade-off between uncertainty quality and the routing framework's overhead. Besides LLM judges, the uncertainty-based router can also send the most uncertain samples to human annotators, potentially augmenting the human preference dataset with previously not covered responses from the policy model. This could help close the gap between reward training and actual policy model distributions.

## Acknowledgments and Disclosure of Funding

We thank Rongzhi Zhang for the discussions at the early phase of our work. We thank the anonymous reviewers for their constructive feedback and suggestions.

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

# A  Chat Templates

We use the messages format and apply the default chat template of Llama-3.1-8B-Instruct[2] to construct concatenated input for pairwise RM. For any prompt, response 1 and response 2, the messages are constructed as follows.

---

**Messages for pairwise comparison**

```
messages = [
    {
        "content": {prompt},
        "role": "user",
    },
    {
        "content": {response_1},
        "role": "assistant_1",
    },
    {
        "content": {response_2},
        "role": "assistant_2",
    },
]
```

---

If the context has multiple turns, we append the messages from `assistant_1` and `assistant_2` after the last message from `user`.

We use the following template as the prompt to request feedback from the LLM judge, which partially uses the HelpSteer2 rubrics for human annotation [52].

---

[2]https://huggingface.co/meta-llama/Meta-Llama-3-8B-Instruct/blob/main/tokenizer_config.json

**Prompt for LLM judge**

```
You are an impartial judge, tasked with evaluating the quality of the two AI
    assistants' responses to the context displayed below.
Your evaluation should be based on the following six criteria:

[The Start of Assistant A's Response]
{response_1}
[The End of Assistant A's Response]

[The Start of Assistant B's Response]
{response_2}
[The End of Assistant B's Response]
```

## B  Policy Gradient Methods

Let $\mathcal{Y}$ denote the response space, then $\pi_{\boldsymbol{\theta}}(\cdot \mid \boldsymbol{x}) \in \Delta(\mathcal{Y})$ admits a probability distribution over $\mathcal{Y}$, and the policy gradient is given by

$$\nabla_{\boldsymbol{\theta}} \mathcal{J}_{\text{policy}}(\boldsymbol{\theta}) = \mathbb{E}_{\boldsymbol{x} \sim \mathcal{D}} \sum_{\boldsymbol{y} \in \mathcal{Y}} \left[ \left( r(\boldsymbol{x}, \boldsymbol{y}) - \beta \log \frac{\pi_{\boldsymbol{\theta}}(\boldsymbol{y} \mid \boldsymbol{x})}{\pi_{\text{ref}}(\boldsymbol{y} \mid \boldsymbol{x})} - \beta \right) \nabla_{\boldsymbol{\theta}} \pi_{\boldsymbol{\theta}}(\boldsymbol{y} \mid \boldsymbol{x}) \right].$$

Since $\pi_{\boldsymbol{\theta}}(\cdot \mid \boldsymbol{x})$ is a probability, it must have $\sum_{\boldsymbol{y} \in \mathcal{Y}} \nabla_{\boldsymbol{\theta}} \pi_{\boldsymbol{\theta}}(\boldsymbol{y} \mid \boldsymbol{x}) = 0$, so we can add any baseline term $b(\boldsymbol{x})$ independent of $\boldsymbol{y}$ to reduce the variance without affecting the exact policy gradient:

$$\nabla_{\boldsymbol{\theta}} \mathcal{J}_{\text{policy}}(\boldsymbol{\theta}) = \mathbb{E}_{\boldsymbol{x} \sim \mathcal{D}, \boldsymbol{y} \sim \pi_{\boldsymbol{\theta}}(\cdot \mid \boldsymbol{x})} [\underbrace{(R(\boldsymbol{x}, \boldsymbol{y}) - b(\boldsymbol{x}))}_{\text{advantage } A(\boldsymbol{x}, \boldsymbol{y})} \nabla_{\boldsymbol{\theta}} \log \pi_{\boldsymbol{\theta}}(\boldsymbol{y} \mid \boldsymbol{x})], \tag{12}$$

where $R(\boldsymbol{x}, \boldsymbol{y}) := r(\boldsymbol{x}, \boldsymbol{y}) - \beta \log \frac{\pi_{\boldsymbol{\theta}}(\boldsymbol{y} \mid \boldsymbol{x})}{\pi_{\text{ref}}(\boldsymbol{y} \mid \boldsymbol{x})}$ denotes the regularized reward. The policy gradient (12) can be estimated by rolling out $\boldsymbol{x} \sim \mathcal{D}$ and $\boldsymbol{y} \sim \pi_{\boldsymbol{\theta}}(\cdot \mid \boldsymbol{x})$, estimating the advantage $\widehat{A}(\boldsymbol{x}, \boldsymbol{y}) = R(\boldsymbol{x}, \boldsymbol{y}) - \widehat{b}(\boldsymbol{x})$ and taking the average. The empirical policy loss at each step is then written as

$$\widehat{\mathcal{L}}_{\text{policy}}(\boldsymbol{\theta}) = -\frac{1}{B} \sum_{i=1}^{B} \widehat{A}(\boldsymbol{x}_i, \boldsymbol{y}_i) \log \pi_{\boldsymbol{\theta}}(\boldsymbol{y}_i \mid \boldsymbol{x}_i). \tag{13}$$

To reduce the variance, the baseline is usually set as the value function (expected reward under the current policy), which is approximated by a critic model as in PPO [39], or by Monte-Carlo (MC) samples as in RLOO [1] and GRPO [43]. In this paper, we focus on the latter approach, which has a more transparent form.

## C  Experiment Details

In this section, we provide additional experimental details about SNGP-PM training, RLOO training and policy evaluations.

### C.1  SNGP-PM Training

Our implementation of SNGP follows [37], particularly the one applied to Bert.[3] Specifically, we only apply spectral normalization to the linear layer in the last decoder and set the spectral normalization range to 1. We set the random feature size $D_r = 4096$, identical to the hidden size $D_h = 4096$ of the Llama-3.1-8B-Instruct model. We set the amplitude $\sigma_k = 1$, scaling factor $\lambda = 10$, and ridge coefficient $\tau = 0.001$. We update all weights during training, except the random feature weights $W$ and $b$ in the GP layer. For the baseline PM, we directly apply a linear layer on the last hidden states of dimension $D_h$ and update all weights during training. For training, we follow the code from OpenRLHF[4], which is an easy-to-use, high-performance open-source RLHF framework [20]. Hyperparameters are summarized in Table 7. PM and SNGP-PM trained with different learning rates are evaluated on HelpSteer2-Preference validation set [51], Reward Bench [25] and RM-Bench [31], and the results are shown in Table 8. We present the results with learning rate 4e-6 in Table 1 as they achieve the highest accuracy on the validation set.

Table 7: Training configurations for PM and SNGP-PM.

| Item | Value |
|---|---|
| Base model name | Llama-3.1-8B-Instruct |
| Batch size | 256 |
| Micro batch size | 16 |
| Training epochs | 2 (3 if counting the covariance calculation pass) |
| Quantization | BFloat16 |
| Learning rate (LR) | {2e-6, 3e-6, 4e-6, 5e-6, 6e-6} |
| Learning rate scheduler | Cosine with min LR (0.1 × base LR) |
| Warm up ratio | 0.03 |
| Gradient accumulation steps | 16 |
| Max input length | 8192 |
| DeepSpeed Zero stage | 2 |
| Flash attention | Enabled |

Table 8: Performance of PM and SNGP-PM trained with varying learning rates.

| Model | LR | Validation (%) | Reward Bench (%) | | | | | RM Bench (%) | | | | |
|---|---|---|---|---|---|---|---|---|---|---|---|---|
| | | acc | avg. | chat | chat hard | safety | reasoning | avg. | chat | math | code | safety |
| PM | 3e-6 | **80.1** | 87.8 | 96.6 | **73.6** | **90.3** | 90.7 | 68.0 | 65.5 | 61.0 | 53.8 | 91.8 |
| | 4e-6 | **80.1** | 87.7 | 96.4 | 73.1 | 89.4 | 91.8 | **68.7** | **67.0** | 60.5 | **55.1** | 92.3 |
| | 5e-6 | 78.7 | **87.9** | **97.8** | 72.1 | 89.1 | 92.5 | 68.4 | 65.3 | **61.5** | 54.8 | 92.1 |
| | 6e-6 | 77.3 | **87.9** | 97.2 | 72.1 | 88.8 | **93.5** | 67.7 | 64.3 | 59.9 | 54.9 | 91.8 |
| SNGP-PM | 2e-6 | 77.8 | 86.8 | 96.1 | 72.5 | 87.9 | 90.7 | 66.9 | 62.9 | **60.6** | 52.6 | 91.5 |
| | 3e-6 | 78.1 | **87.1** | 95.8 | 71.2 | 89.2 | 92.0 | **68.1** | 65.7 | 60.1 | 53.3 | **93.1** |
| | 4e-6 | **79.3** | **87.3** | 95.8 | **73.8** | **89.4** | 90.0 | 68.0 | **67.1** | 59.5 | **54.2** | 91.2 |
| | 5e-6 | 76.7 | 86.3 | **96.6** | 70.1 | 86.3 | **92.1** | 67.4 | 64.0 | 58.8 | 53.9 | 92.9 |

### C.2  RLOO Training

For the Ultrafeedback prompt dataset, we extract the prompts from the preference version[5] used in [21]. Given resource constraints, we sample the first 33% of the dataset, which consists of 19,456 prompts covering a wide range of domains. Our implementation of RLOO follows OpenRLHF

---

[3]https://github.com/google/uncertainty-baselines/blob/main/baselines/clinc_intent/sngp.py

[4]https://github.com/OpenRLHF/OpenRLHF

[5]https://huggingface.co/datasets/allenai/tulu-2.5-preference-data/viewer/default/ultrafeedback_overall

[20], with modifications serving the SNGP-PM with router instead of the standard pointwise RM. In particular, we move the advantage computation from the trainer to the PM side. We run experiments on 8×NVIDIA A100-80G GPUs, using 2 GPUs serving the PMs in parallel, co-locating the actor and reference models on 4 GPUs, and using the remaining 2 GPUs for on-policy sampling. For LLM-as-a-judge, we call a remotely hosted DeepSeek-R1, which allows 200 requests per minute. We use the prompt template in Appendix A, and convert the returned label to a reward difference in $\{-2, 0, 2\}$. Training hyperparameters are summarized in Table 9.

Table 9: Training configurations for RLOO.

| Item | Value |
|------|-------|
| Base model name | Llama-3.1-8B-Instruct |
| Rollout batch size | 1024 |
| Train batch size | 1024 |
| Micro rollout batch size | 128 |
| Micro train batch size | 8 |
| Training episodes | 1 |
| Max prompt length | 2048 |
| Max generation length | 1024 |
| Quantization | BFloat16 |
| Actor learning rate | 5e-7 |
| KL coefficient | 0.01 |
| Advantage clip ratio | 0.2 |
| Samples per prompt ($K$ in RLOO) | 4 |
| DeepSpeed Zero stage | 3 |
| Flash attention | Enabled |

## C.3 Policy Model Evaluations

For Arena-Hard-v0.1 [26], we use the official library,[6] adopting the default decoding configuration and comparing the WR against GPT-4-0314, using GPT-4.1 as the judge. For AlpacaEval 2.0 [12], we follow the default setting,[7] evaluating the WR against GPT-4-Turbo using weighted_alpaca_eval_gpt4_turbo as annotator. When generating the output, we use the default generation configuration of Llama-3.1-8B-Instruct.[8]. We run the evaluation 3 times and report the average WR and length-controlled (LC) WR. For MT-Bench [62], we follow an open codebase[9] and update the chat format for compatibility with the Llama-3.1-8B-Instruct chat template. We use GPT-4-Turbo as the judge to rate the quality of responses with scalar scores ranging from 1 to 10. Detailed results on Arena-Hard-v0.1 and AlpacaEval 2.0 are provided in Tables 10 and 11.

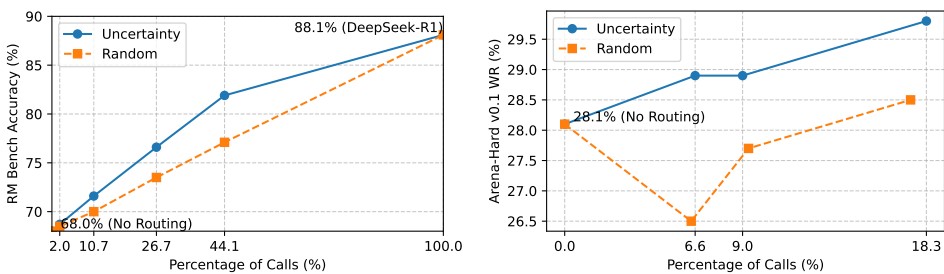

Figure 3: Uncertainty-based routing outperforms random routing with the same number of strong LLM judge calls on preference accuracy (RM-Bench) and downstream alignment (Arena-Hard-v0.1).

---

[6]https://github.com/lmarena/arena-hard-auto

[7]https://github.com/tatsu-lab/alpaca_eval

[8]https://huggingface.co/meta-llama/Llama-3.1-8B-Instruct/blob/main/generation_config.json

[9]https://github.com/fanqiwan/FuseAI/tree/main/FuseChat-3.0/FuseEval/IF-Eval/MT-Bench

Table 10: Performance comparison of downstream policy models on Arena-Hard-v0.1 with GPT-4.1 as judge. WR stands for win rate against GPT-4-0314, and CI stands for confidence interval.

| Model | Num of Calls | Arena-Hard-v0.1 (%) | |
| --- | --- | --- | --- |
| | | WR | CI |
| Base model | - | 24.5 | (-1.8 / +1.4) |
| No routing | 0 | 28.1 | (-1.9 / +1.6) |
| Uncertainty (1.35) | 7668 (6.56%) | **28.9** (+2.4) | (-2.4 / +1.7) |
| Uncertainty (1.30) | 10522 (9.01%) | **28.9** (+1.2) | (-2.0 / +2.0) |
| Uncertainty (1.20) | 21363 (18.3%) | **29.8** (+1.3) | (-2.2 / +1.9) |
| Random (1.35) | 7523 (6.40%) | 26.5 | (-1.5 / +1.5) |
| Random (1.30) | 10854 (9.30%) | 27.7 | (-1.8 / +1.9) |
| Random (1.20) | 20474 (17.5%) | 28.5 | (-2.0 / +1.8) |

Table 11: Performance comparison of downstream policy models on AlpacaEval 2.0 with GPT-4 Turbo as judge. LCWR is the length-controlled win rate, and WR is the win rate. Avg Length shows average generation length.

| Model | Num of Calls | AlpacaEval 2.0 | | |
| --- | --- | --- | --- | --- |
| | | LCWR (%) | WR (%) | Avg. Length |
| Base model | - | 22.31 | 23.63 | 2304 |
| No routing | 0 | 25.40 | 27.35 | 2142 |
| Uncertainty (1.35) | 7668 (6.56%) | **26.28** (+0.58) | **28.97** (+0.42) | 2133 |
| Uncertainty (1.30) | 10522 (9.01%) | **26.34** (+1.20) | **28.53** (+0.24) | 2163 |
| Uncertainty (1.20) | 21363 (18.3%) | **26.45** (+0.47) | **28.91** (+0.40) | 2167 |
| Random (1.35) | 7523 (6.40%) | 25.70 | 28.55 | 2085 |
| Random (1.30) | 10854 (9.30%) | 25.14 | 28.29 | 2157 |
| Random (1.20) | 20474 (17.5%) | 25.98 | 28.51 | 2189 |

## C.4 Judge Latency

The judge latency depends on model and its serving efficiency, as well as other engineering factors in the pipeline. In our experiment, we use DeepSeek-R1 through API calls, which is hosted and allows 200 requests per minute (RPM). In our RLHF experiment, the batch size is 256, and we generate 4 samples per prompt, resulting in 1536 total pairwise comparisons per batch, and only about 6% to 9% to 18% of them are routed to the judge (see Table 6), corresponding to 92 to 138 to 276 judge requests per batch. Therefore, most of the requests within a batch can be executed in complete parallel (no backlog), and the overhead is further reduced given that requests are sent asynchronously. Notably, our use of 200 RPM-limited DeepSeek-R1 is due to the early time when experiments are conducted and the budget constraints. In current practice, various strong LLMs can be used, possibly with a higher rate limit. In a practical online RLHF setting, the routing framework can be further optimized by dynamically adjusting the uncertainty threshold to ensure the number of judge requests stays within the limit, preventing the pipeline from blocking.

