# OpenReview forum: "Ask a Strong LLM Judge when Your Reward Model is Uncertain"
_NeurIPS.cc/2025/Conference — NeurIPS 2025 poster_

### Official Review · Reviewer_Hyxs · 2025-07-02

**Clarity:** 3
**Significance:** 2
**Originality:** 2
**Rating:** 4
**Confidence:** 4

**Summary:**

This paper is on reward models (RMs) in the context of reinforcement learning for fine-tuning large language models. The main contribution is an uncertainty-based routing framework designed to improve the efficiency and quality of the fine-tuning process. The proposed method evaluates the uncertainty of its internal reward model. When the RM's confidence is low on a given sample, the framework defers the decision to a more powerful, albeit costly, external LLM judge. This dynamic allocation aims to leverage the high-quality signal from the stronger judge only when necessary, thus optimizing the trade-off between cost and performance. The authors validate their approach by applying the routing strategy to fine-tune a Llama 8B model, demonstrating its effectiveness empirically.

**Questions:**

The paper's main contribution is an uncertainty-based routing strategy, which I find promising.
My primary questions revolve around including more UQ methods, as this could further strengthen the paper's claims.
The framework's effectiveness is fundamentally tied to the reliability of its uncertainty estimates.

- Have the authors considered or experimented with other well-established and computationally inexpensive UQ techniques for the reward model? A broader comparison could provide a more robust foundation for the paper.

Possible UQ techniques include:
- MC Dropout: a relatively simple yet powerful technique to implement, especially for the classification head of the reward model. It would be an interesting and straightforward comparison.
- Laplace Approximation: A last-layer Laplace approximation is another strong candidate for this use case. Since the final classification head likely has a manageable number of parameters (d×1), a full Laplace approximation on this layer might even be feasible and could provide a more principled uncertainty estimate. Could the authors comment on the potential of using such a method within their framework?

**Ethical Concerns:**

["NO or VERY MINOR ethics concerns only"]

**Final Justification:**

I thank the author for the rebuttal. I maintain the positive score.

**Limitations:**

yes

**Quality:**

3

**Strengths And Weaknesses:**

Strengths:
- The core concept of the paper is well-motivated and intuitive. Using model uncertainty to dynamically route queries between a local, efficient RM and a more capable external judge is a logical approach to balancing signal quality and computational cost.
- The work provides a valuable direction for future research. While not the first to use uncertainty quantification, its application as an explicit routing mechanism in a hybrid system is a practical and interesting contribution that could inspire further work in efficient LLM alignment.
- The empirical results effectively demonstrate the value of the proposed method. The comparison against a random routing baseline clearly shows that the uncertainty-based strategy is superior, making a case for its adoption.

Weaknesses:
- The exploration of different uncertainty quantification (UQ) techniques is limited to a single method. Given that the performance of the entire framework hinges on the quality of the uncertainty estimates, a more thorough investigation of alternative UQ methods would be beneficial. This is also because there is a vast literature on UQ techniques for neural networks that can be used.
- The scope of the experimental validation is somewhat limited. The experiments are conducted only on a Llama 8B model; while this is a significant undertaking, demonstrating the framework's effectiveness on even larger, state-of-the-art models would make the conclusions more generalizable.

---

> ### Author Rebuttal · Authors · 2025-07-31
>
> We appreciate the reviewer for the positive feedback and highlighting the promising nature of our uncertainty-based routing strategy. We address their concerns in the following.
>
> > W1&Q1: The exploration of different uncertainty quantification (UQ) techniques is limited to a single method. Given that the performance of the entire framework hinges on the quality of the uncertainty estimates, a more thorough investigation of alternative UQ methods would be beneficial. This is also because there is a vast literature on UQ techniques for neural networks that can be used.
> > Have the authors considered or experimented with other well-established and computationally inexpensive UQ techniques for the reward model? A broader comparison could provide a more robust foundation for the paper?
>
> Thank you for raising this concern/question, and we agree that a broader comparison of UQ techniques is an important research direction. That said, we would like to highlight that our main contribution is introducing the uncertainty routing framework for the reward model, enabling more reliable reward signals and better online RLHF in a cost-efficient way. Within this framework, various UQ methods can be applied, and we believe a good UQ method can potentially lead to better performance.
>
> With budget constraint and limited capacity (we conduct experiments not only on reward benchmarks but also downstream alignment), we are not able to explore many other UQ methods. We chose SNGP for proof-of-concept because of its: (1) low inference cost; (2) implementation simplicity; (3) principled design. Firstly, SNGP only requires a single forward pass at inference time for both prediction and uncertainty score, which is a desired property as an RM must evaluate many response pairs quickly in the online RLHF pipeline. Secondly, its implementation is relatively simple, and it does not require much modification to the standard reward learning pipeline. Thirdly, it is designed to be distance-aware, making it particularly well-suited for detecting OOD samples that are the central challenge for reliable reward.
>
> Regarding the UQ methods mentioned in the question:
>
> - MC Dropout: We agree that MC Dropout is a straightforward and powerful technique, and we tried it in the early stage to examine the relation between accuracy and uncertainty. However, it requires multiple forward passes at inference time to estimate uncertainty. In our setting, this would multiply the inference cost substantially, even without counting the judge calls, creating a latency bottleneck that conflicts with the efficiency requirements of an online RLHF pipeline. Therefore, we view it as less suitable for our framework compared to a single-pass method like SNGP.
> - Laplace Approximation: Last-layer Laplace approximation is closely related to our method, as SNGP also implements Laplace approximation. As detailed in the original SNGP paper [1], the method replaces the final dense layer with an approximate Gaussian Process and computes the posterior using the Laplace approximation on its random feature expansion. This provides a principled Bayesian perspective that yields a closed-form posterior, which can be learned scalably with minimal modification to the standard training pipeline. While a direct Laplace approximation on the final layer is another valid approach, our chosen method, SNGP, allows us to scale similarly efficiently while leveraging strong theoretical properties, serving as a good baseline.
>
> > W2: The scope of the experimental validation is somewhat limited. The experiments are conducted only on a Llama 8B model; while this is a significant undertaking, demonstrating the framework's effectiveness on even larger, state-of-the-art models would make the conclusions more generalizable.
>
> Firstly, we would like to highlight that our experiment is not limited to testing reward accuracy on benchmarks (Reward Bench and RM-Bench) but also includes the improvement brought to downstream policy model alignment, which makes our validation rather comprehensive.
>
> We agree that demonstrating the framework's effectiveness on larger models is an important step for generalization. Our experiments, which involve computationally intensive online RLHF, were designed to provide a robust proof-of-concept for the uncertainty-routing framework. We chose the Llama-3.1-8B model as a representative foundation to establish this initial validation within the budget constraint. We consider scaling these experiments to larger models for future work.
>
> [1] Liu, Jeremiah Zhe, et al. "A simple approach to improve single-model deep uncertainty via distance-awareness." JMLR 2023.

---

### Official Review · Reviewer_Qbyt · 2025-07-03

**Clarity:** 3
**Significance:** 2
**Originality:** 2
**Rating:** 4
**Confidence:** 4

**Summary:**

This paper proposes an uncertainty based scheme that queries a strong LLM to label preference in RL/PG training of LLM. The algorithm first trains a pairwise RM and estimates its uncertainty using GP. At policy training time, if the uncertainty on a given pair of trajectories (of a batch sampled by the policy) is higher than a threshold, then the strong model is queried to provide a label; otherwise, the learned RM is used. The authors show the proposed improves the reward prediction quality and is more efficient than random routing. In addition, they show policy learning from this LLM-assisted reward improves the policy performance.

**Questions:**

1. In Eq (8) why is p(x,y_1,y_2) defined as g/u instead of g?

2. How are the hyperparameters of SNGP part chosen?

3. How much more tokens does the Uncertainty-based method use than Random routing in the LLM training experiments? In the previous RM experiments, it suggest Uncertainty-based generates more tokens.

4. The current highlights in Table 6 is a bit misleading. e.g. in MT-Bench Random (1.35) performs on-part with Uncertainty (1.3) on Avg and better than the rest except No routing on Turn 1. I would suggest highlighting just one row per column.

5. Can you explain why the improvement of proposed methods vary a lot from different benchmarks in Table 6? E.g. how does it relate to the ability of LLM to judge? In particular, in Arena-Hard and MT-bench, the results are not much better than the baseline of no-routing, especially given the noises, I'm not sure if the current results are statistically significant.

6. Please add a discussion on literature of active learning, especially in the context of RL.

**Ethical Concerns:**

["NO or VERY MINOR ethics concerns only"]

**Limitations:**

Yes

**Quality:**

3

**Strengths And Weaknesses:**

S1. The paper is well motivated and RLAIF or RL with LLM is an important area now.

S2. The proposed algorithm seems straightforward and scalable. The current experimental results seem promising.

S3. The paper is overall well written.

S4. The paper reveals a good insight on the difference between pointwise and pairwise RM in uncertainty estimation,

W1. The paper lacks discussion of the active learning literature, which has a large corpus of similar works, and some are for RL, though not necessarily for LLM training.

W2 The improvement is LLM training is not super significant.

---

> ### Author Rebuttal · Authors · 2025-07-31
>
> We appreciate the reviewer for the positive feedback and for acknowledging the strengths of our work, and we would like to answer the questions in the following.
>
> > Q1: In Eq (8) why is $p(x,y_1,y_2)$ defined as $g/u$ instead of $g$?
>
> Eq (8) comes from the mean-field approximation to the Gaussian posterior, as presented in Eq (19) in the original SNGP paper [1]. Intuitively, the preference score $p$ encompasses both aleatoric and epistemic uncertainties, while $g$ only conveys in-distribution aleatoric uncertainty through its scale. Thus, by dividing $u$ conveying epistemic uncertainty, the resulting score $p$ is less overconfident on OOD samples.
>
> > Q2: How are the hyperparameters of SNGP part chosen?
>
> The SNGP hyperparameters are chosen based on the recommended tuning strategy in [1] (Appendix A.2). In particular, we choose the spectral norm bound ensuring that the validation accuracy remains close to vanilla PM. The GP dimension is chosen to match the model's (Llama-3.1-8B-Instruct) hidden dimension. Kernel amplitude and scaling factors do not affect accuracy but only impact the range of scores and uncertainties; thus, we tune them so that the scores' range matches vanilla PM and choose the uncertainty router threshold accordingly. Ridge coefficient is insignificant in our experiments, so we just set it as recommended in [1].
>
> > Q3: How much more tokens does the Uncertainty-based method use than Random routing in the LLM training experiments?
>
> We thank the reviewer for this good question. While uncertainty-based routing consumes more tokens for the same number of routed pairs, it actually uses *less* tokens to achieve the same prediction accuracy. In particular, by combining tables 2 and 3 (also 4 and 5), we can plot accuracy vs inference time curves and observe that uncertainty-based routing uses approximately 55% less time than random routing to achieve 89.1% accuracy on RewardBench, and approximately 16% less time to achieve 77.1% accuracy on RMBench in our setting. Thus, given that the number of tokens scales with inference time, our experiments suggest that fewer tokens are required by uncertainty-based routing.
>
> > Q4: The current highlights in Table 6 is a bit misleading. I would suggest highlighting just one row per column.
>
> Thank you for the suggestion, and we will revise it accordingly, with one highlighting row per column.
>
> > Q5: Can you explain why the improvement of proposed methods vary a lot from different benchmarks in Table 6? E.g. how does it relate to the ability of LLM to judge? In particular, in Arena-Hard and MT-bench, the results are not much better than the baseline of no-routing, especially given the noises, I'm not sure if the current results are statistically significant.
>
> Thank you for the question. Firstly, we would like to highlight that the benchmarks in Table 6 have different focuses.
>
> - Arena-Hard focuses on challenging user prompts (open-ended software engineering problems, math questions, etc) where the win rate (WR) is the metric.
> - AlpacaEval focuses on general helpfulness and human instruction following, where the length-controlled win rate (LCWR) is more important, as the naive WR may possess length bias and has less correlation with real-world human preference.
> - MT-Bench focuses on testing the multi-turn dialog ability, where the Turn 2 score (also average score) is more important.
>
> Given the nature of these benchmarks, we aim to compare our uncertainty-routing method with no/random routing mainly on the most important metrics, namely Arena-Hard WR, AlpacaEval LCWR, and Turn 2 score in MT Bench. On all of these 3 metrics, we observe consistent improvement over the baselines, thus showing the efficacy of our method.
>
> Regarding the performance variation, we attribute it to the following factors:
>
> 1) Benchmark and judge mismatch: Benchmark metrics reflect different, sometimes even conflicting skills, and the evaluation judges (like GPT-4.1 and GPT-4-Turbo) differ from the DeepSeek-R1 judge used during training, creating variance.
> 2) Multi-objective trade-offs: On MT-Bench, our method slightly lowers the Turn 1 score but improves the more difficult Turn 2 score, leading to a better conversational model.
>
>
> > Q6: The paper lacks discussion of the active learning literature.
>
> We thank the reviewer for this valuable suggestion. We agree that our work has strong connections to active learning and that a more detailed discussion would better contextualize our contributions. In Section 5, we briefly allude to this by suggesting that uncertain samples could be sent to human annotators to augment the training data.
> To make this connection explicit, we will expand this discussion in the revision with the following paragraph:
>
> "Our uncertainty-based routing framework is conceptually aligned with active learning, which seeks to improve data efficiency by strategically selecting the most informative instances for labeling by an oracle [4]. In our work, the strong LLM judge acts as the oracle, and our uncertainty score serves as the acquisition function to identify the most uncertain pairs of responses. Classic query strategies in active learning often rely on model uncertainty, such as selecting the least confident predictions or using a committee of models to find contentious examples [5], which have been extended to deep learning through Bayesian methods [6]. While active learning has been explored for learning reward/preference functions from limited interaction in the RL context [7,8], our primary contribution is the application of this principle to a direct online RLHF setting. Here, the feedback from the oracle is used immediately to construct advantage estimates for policy gradient updates, and feeding these newly labeled high-uncertainty pairs back into the reward model for continued training is a natural extension that would constitute a full active learning cycle."
>
>
> [1] Liu, Jeremiah Zhe, et al. "A simple approach to improve single-model deep uncertainty via distance-awareness." JMLR 2023.
>
> [2] Li, Tianle, et al. "From crowdsourced data to high-quality benchmarks: Arena-hard and benchbuilder pipeline." arXiv preprint arXiv:2406.11939 (2024).
>
> [3] Dubois, Yann, et al. "Length-controlled alpacaeval: A simple way to debias automatic evaluators." arXiv preprint arXiv:2404.04475 (2024).
>
> [4] Settles, Burr. "Active learning literature survey." (2009).
>
> [5] Seung, H. Sebastian, Manfred Opper, and Haim Sompolinsky. "Query by committee." COLT 1992.
>
> [6] Gal, Yarin, Riashat Islam, and Zoubin Ghahramani. "Deep bayesian active learning with image data." ICML 2017.
>
> [7] Houlsby, Neil, et al. "Bayesian active learning for classification and preference learning." arXiv preprint arXiv:1112.5745 (2011)..
>
> [8] Daniel, Christian, et al. "Active Reward Learning." Robotics: Science and systems. Vol. 98. 2014.

---

### Official Review · Reviewer_WJxQ · 2025-07-03

**Clarity:** 4
**Significance:** 3
**Originality:** 2
**Rating:** 5
**Confidence:** 3

**Summary:**

This paper introduces an uncertainty-based routing framework for improving reward model (RM) performance in RLHF settings. The central idea is to quantify the RM’s uncertainty using a pairwise preference model (PM) augmented with Spectral-normalized Gaussian Process (SNGP) uncertainty estimation. When the RM is uncertain about a pairwise comparison, the sample is routed to the more reliable but slower LLM judge; otherwise, the RM’s output is used directly. This system is evaluated on standard reward modeling benchmarks and downstream alignment tasks, showing that routing based on uncertainty leads to higher accuracy and improved policy alignment compared to random routing under similar resource constraints.

**Questions:**

See weaknesses plz.

**Ethical Concerns:**

["NO or VERY MINOR ethics concerns only"]

**Final Justification:**

Considering the comments of other reviewers, I maintain my positive rating. But I would like to increase the Clarity score by one point. I hope that the revised version will include the discussion in Section 5 and the Appendix, as the authors have promised.

**Limitations:**

Yes.

**Paper Formatting Concerns:**

No.

**Quality:**

3

**Strengths And Weaknesses:**

Strengths: The topic is aligned with the conference community. This paper has solid technical motivations and effective methods for routing. It has comprehensive experimental evaluation, which makes the conclusion persuasive.
 This paper briefly acknowledges the limitations and proposes reasonable future improvements.

Weaknesses:

•	Inadequate Analysis of Judge Latency in Large-Scale RLHF: While cost and parallelism are mentioned (Tables 3 and 5), there isn't a detailed operational account of potential delays in practical large-scale RLHF settings where many judge calls may backlog.

•	Limited Diversity in LLM Judge Usage: The limited diversity in LLM judge usage is a notable aspect of this work. All experiments rely on DeepSeek-R1, which, as the authors acknowledge, is not specifically optimized for judging and requires significant resources. I wonder if the authors explored the potential of utilizing other LLMs, particularly lightweight models, as judges. While the absolute performance might not match DeepSeek-R1, exploring the trade-off between effectiveness and efficiency could yield valuable insights and would be a worthwhile addition to the experimental analysis.

---

> ### Author Rebuttal · Authors · 2025-07-31
>
> We appreciate the reviewer for their positive assessment and insightful feedback. We address the Weaknesses below.
>
> > W1:  Inadequate Analysis of Judge Latency in Large-Scale RLHF: While cost and parallelism are mentioned (Tables 3 and 5), there isn't a detailed operational account of potential delays in practical large-scale RLHF settings where many judge calls may backlog.
>
> Thank you for this question. While our experiments in Tables 3 and 5 aimed to provide a comparative analysis of the time cost between uncertainty and random routing strategies, we agree that a more detailed operational account of managing judge latency is a valuable addition.
>
> In particular, the latency depends on which judge is used and how it is served, as well as other engineering factors in the pipeline. In our experiment, we use DeepSeek-R1 through api calls, which is hosted by a cloud provider that allows 200 requests per minute (RPM), and it can take from 20 seconds to above 1 minute to generate an output depending on the reasoning length. In our RLHF experiment, the batch size is 256, and we generate 4 samples per prompt, resulting in 1536 total pairwise comparisons per batch, and only about 6% to 9% to 18% of them are routed to the judge (see Table 6), corresponding to 92 to 138 to 276 judge requests per batch. Therefore, most of the requests within a batch can be executed in complete parallel (no backlog), and the overhead is further reduced given that requests are sent and completed asynchronously.
>
> Notably, our use of 200 RPM limited DeepSeek-R1 is due to the early time when experiments are conducted and the budget constraints. In current practice, there are various strong LLMs that can be used, possibly with a higher rate limit. For example, tier 1 user can have 500 RPM for GPT o3/o3-pro and 1000 RPM for o3-mini.
>
> In a practical online RLHF setting, the routing framework can be further optimized by dynamically adjusting the uncertainty threshold $u$ to ensure the number of judge requests stays within the limit, preventing the pipeline from blocking.
> We will add a discussion of these operational considerations to the appendix to provide a clearer picture for practitioners looking to implement this framework.
>
> > W2: Limited Diversity in LLM Judge Usage: The limited diversity in LLM judge usage is a notable aspect of this work. All experiments rely on DeepSeek-R1, which, as the authors acknowledge, is not specifically optimized for judging and requires significant resources. I wonder if the authors explored the potential of utilizing other LLMs, particularly lightweight models, as judges. While the absolute performance might not match DeepSeek-R1, exploring the trade-off between effectiveness and efficiency could yield valuable insights and would be a worthwhile addition to the experimental analysis.
>
> We thank the reviewer for raising this point. We would like to clarify that our uncertainty-based routing strategy is generic and can be applied to any LLM judge. In particular, our judge prompt in Appendix B is not specifically tailored for any specific LLM. Our choice of DeepSeek-R1 is due to its high capability within our budget constraint. We believe that stronger models and specialized judges can be used to improve the reward signals.
>
> Nevertheless, the primary goal of our work is to establish a strong proof-of-concept for the uncertainty-routing framework. By using the powerful, high-accuracy R1 judge, we measure the potential of improvement one could gain from routing to a strong oracle. If the framework showed no significant benefit with a powerful judge, its utility would be questionable.
>
> Having established this viability, exploring more resource-efficient judges is a logical next step for future work. As briefly noted in our conclusion, one could replace it with a smaller-scale generative RM to further improve the judge quality and inference efficiency. This opens up the possibility of a hierarchical routing system, where a local fast RM handles the most certain pairs, a lightweight, efficient judge (e.g., a fine-tuned 2B generative RM) handles cases of moderate uncertainty, and an expensive but powerful judge is reserved for only the most uncertain and critical cases. This approach would offer a more nuanced balance between cost, latency, and feedback quality. We will expand this discussion in Section 5 to highlight the potential of diverse LLM judge usage.

---

> ### Author Response · Authors · 2025-08-06
>
> Dear Reviewer WJxQ,
>
> Thank you again for your effort reviewing our paper, providing positive assessment and valuable feedback. Since the discussion period is ending soon, we would like to follow up to see if our rebuttal has addressed your concerns and if there are any other questions. We would be happy to provide any further clarification. Thank you very much.
>
> Sincerely,
>
> Authors of Submission 22771

---

> > ### Comment · Reviewer_WJxQ · 2025-08-07
> >
> > Thank you for the rebuttal, especially for the analysis of potential delays. We understand that our concerns do not affect the main contributions of the article and appreciate the authors' efforts to provide as detailed explanations as possible within the limited time available.
> >
> > Considering the comments of other reviewers, we maintain our positive rating. But we would like to increase the Clarity score by one point. We hope that the revised version will include the discussion in Section 5 and the Appendix, as the authors have promised.

---

> > > ### Author Response · Authors · 2025-08-07
> > >
> > > Thank you for your positive feedback, and we will include our discussion in the revision as promised in the rebuttal.

---

### Note · Authors · 2025-08-14

We sincerely thank all reviewers for their constructive feedback and engaging discussion. We are encouraged by the consensus that our uncertainty-based routing framework is a well-motivated, intuitive, and practical approach to improving the reward quality while maintaining a reasonable cost for better downstream online RLHF.

The discussions were highly productive. We are pleased that our rebuttal addressed the concerns of Reviewer WJxQ, who subsequently raised their Clarity score, and Reviewer Qbyt, who maintained their positive rating. Based on this valuable feedback, we will incorporate the following key revisions to strengthen the final manuscript:

- Expanding context and future work: We will add the promised discussion on the connections to active learning (Qbyt), elaborate on the practical operational latency of LLM judges in the appendix (WJxQ), and expand our conclusion to discuss the potential for hierarchical routing with diverse judges and other UQ methods as avenues for future work (Hyxs).
- Improving clarity and presentation: We will revise our tables and figures for maximum clarity, as suggested by Reviewer Qbyt.

We believe these revisions address the points raised by reviewers and will further strengthen our paper's contributions and improve their clarity for the community. We thank the AC for their time and consideration.

---

### Decision · Program_Chairs · 2025-09-17

**Decision:**

Accept (poster)

**Comment:**

This paper proposes that a lightweight reward model (in RLHF) be equipped with an uncertaintly quantification method that decides when the RM is uncertain about a particular generation and therefore sends it to a larger/stronger model for judgement. The method is shown to improve over equivalent baselines in downstream policy effectiveness.

Reviewers agreed that the approach is an elegant and principled way to solve the problem of doing RLHF while optimizing resources. The topic is a very important one as it addresses the RLHF pipeline that is ubiqutious in industry. The experimental results show positive gains, but reviewers did point out a few different dimensions that ideally would have been explored more (e.g. different UQ methods, different Judge LM's etc), but these experiments would be expensive so maybe out of scope. A larger concern I have personally is that measuring success through # of calls to the RM may not be a perfect measure of efficiency since an RM equipped with a UQ method (like SNGP) will be somewhat more FLOP-intensive. iso-flops measurements may have been more helpful.

Still, all reviewers found the contribution good and it is worth acceptance.